# Automated Dimension Estimation of Steel Pipes Stacked at Construction Sites Using Euclidean Distances Calculated by Overlapping Segmentation

**DOI:** 10.3390/s22124517

**Published:** 2022-06-15

**Authors:** Yoon-Soo Shin, Junhee Kim

**Affiliations:** Department of Architecture, College of Engineering, Dankook University, 152 Jukjeon-ro, Yongin-si 16890, Gyeonggi-do, Korea; shinys@dankook.ac.kr

**Keywords:** materials management, image processing, convolutional neural network, perspective transformation, image segmentation, construction

## Abstract

Pipes are construction materials for water and sewage, air conditioning, firefighting, and gas facilities at construction sites. The quantification and identification of pipes stacked at construction sites are indispensable and, thus, are directly related to efficient process management. In this study, an automated CNN-based technique for estimating the diameter and thickness of the pipe in an image is proposed. The proposed method infers the thickness of the pipe through the difference by segmentation, by overlapping the inside and outside circles for a single pipe. When multiple pipes are included in the image, the inside and outside circles for the identical pipe are matched through the spatial Euclidean distance. The CNN models are trained using pipe images of various sizes to segment the pipe circles. An error of less than 7.8% for the outer diameter and 15% for the thickness is verified through execution with a series of 50 testing pipe images.

## 1. Introduction

Various types of building materials are used at the construction sites of structures, which are aggregates of complex engineering systems. In a construction project, material management is essential because it is directly related to an efficient process control [1]. The main task of material management is to identify the types, sizes, and quantities of materials brought into and out of the construction site. The management has been traditionally conducted by workers, however, it is rather subjective, time-consuming, dangerous, and error-prone. Therefore, as an alternative to the conventional manual material management, an automated material management that minimizes manpower has been suggested [2]. Various automated material management systems are leveraged by laser and image-based technologies that target construction materials, e.g., rebar, brick, and pipe, which are mainly small in size and consumed in large amounts.

A variety of laser equipment, which has been disseminated from laser-based technologies of autonomous vehicles and virtual reality systems, has been applied to material management technology at construction sites [3,4]. A laser triangulation of the phase-change detection of the line lasers is applied to detect the cross-section and side surfaces, for both the stacked rebars and the individual rebars moving on a conveyor [5]. Kinect, a laser device that emits multiple laser sources from a single sensor, generates a point cloud in a three-dimensional space and thus measures the size of the target objects directly [6]. A study on the size and the counting of rebars [7], and a study on measuring the diameter of pipes [8,9] using the Kinect, were conducted, respectively.

The image-processing technique has gathered attention at construction sites where a large number of workers work in complex spaces, due to the advantage that it can be combined with mobile devices without the installation of additional hardware equipment [10]. A study was conducted to recognize individual rebars by combining a cross-section and top view image of the rebars [11]. The edge of the cross-section of rebars was distinguished through the pixel values according to the contrast change of the cross-section image of the stacked rebars [12]. The number and pattern of the bricks uniformly installed on a plane were recognized to determine the quantity of bricks [13]. Image processing has its limitations in recognition accuracy, depending on the environmental variables, such as luminance, shading, the angle at which the image was acquired, and the shape of an object, such as the material’s size, overlap, and occlusion. However, object recognition accuracy has rapidly increased with the recent development of convolutional neural network (CNN)-based deep learning [14,15].

A neural network model, generated by the CNN, is composed of a convolution filter that discriminates image features through repetitive training [16]. The model, trained with a sufficient number of images and repetitions, shows a high recognition rate regardless of the image acquisition environment and the shape of the object and, thus, has been applied to various material management studies. A high recognition rate, regardless of the heterogeneous material, size, shape, and the arrangement of bricks, was reported for a brick-recognition method using a CNN [17]. In addition, a high recognition rate, regardless of the occurrence of rust or overlapping with adjacent rebars, was achieved [18]. Pipes with a cross-section similar to that of the rebar were counted as individual pipes in the dense pipe pile [19]. Since pixels in the image can be replaced by distance units, a study on the diameter estimation of rebars was conducted [20].

Pipes, which are essential elements for water and sewage, air conditioning, firefighting, and gas facilities, are one of the most common construction materials at construction sites [21]. However, a limited volume of research has been conducted on construction material management [22]. In this study, an automated CNN-based dimension estimation of pipes is developed to estimate the thickness along with the diameter of pipes to cope with various types of pipes. The diameter and thickness of a pipe were estimated through an overlapping segmentation of the outer and inner diameters of the pipe. When multiple pipes are included in a single image, the outer and inner circles of the referred pipe are matched through the spatial Euclidean distance. In addition, a perspective transformation on the image is performed through homography by utilizing a tag attached to the pipe bundle. The paper is organized as follows: the dataset configuration for CNN model training and the CNN-based image-processing techniques for estimating the diameter and thickness of pipes are explained in Section 2. The generation of the CNN models and an evaluation of the sample images are followed in Section 3. The paper concludes with a brief summary and discussions in Section 4.

## 2. Development of Pipe Diameter and Thickness Estimation

### 2.1. Research Method

In this study, an automated method based on image processing to estimate and measure the pipe diameter and thickness was developed as a means to monitor the automatic import and export of pipes at construction sites. The recent continuous development of the CNN algorithms in the field of image processing has enhanced the performance of object recognition in images. In particular, a single camera can be advantageously used to manage all the materials when several materials of similar shapes and sizes are stacked at the construction site.

Pipe counting and their size estimation within an image can enable to infer the diameter of a pipe from the cross-section detection and the pixel range occupied by the cross-section of the individual pipes in the image; this can be obtained from a pre-acquired dataset through segmentation from a CNN model that has been applied with supervised learning. In addition, a homography based on image processing converts the pixel coordinates into displacement coordinates while simultaneously converting the perspective-view images to align the images of the front view, meaning that normalization is enabled and the acquirement of the pipe diameter that reflects the actual scale becomes possible. However, as pipes of the same diameter can have varying thicknesses (e.g., for a nominal pipe size [23], according to the United States standard for pipe sizes, there are 12 thicknesses in the case of NPS8, which has an outside diameter of 8.625 in), the thickness must additionally be inferred along with the size of the pipe. The thickness of a pipe refers to the closest linear distance where the points from both the outside circle and the inside circle of the pipe simultaneously satisfy verticality, thus indicating that the thickness can be inferred from the difference between the diameters of the outside and inside circles on the same line from the center of the pipe.

The estimation of the pipe diameter and thickness consists of three major sequential algorithms. As shown in Figure 1, the first step is the creation of a CNN model by using the images entailing various pipe cross-sections for object detection and instance segmentation. The dataset consists of a training dataset, a validation dataset for preventing overfitting, and a test dataset for verifying the trained model, and an optimal CNN model is derived from the k-fold validation test. If an image is entered into the CNN model that can segment objects into polygons through the feature extraction and region proposal processes, the object becomes segmented into sections of the outside circle, the inside circle, and the name tag. The second process is to deduce the transformation matrix from the name tag of the segmented image and apply homography to the entire image to acquire an aligned image of the front view, which reflects the actual scale. Finally, the Euclidean distance is calculated to match the outside circle and the inside circle of the identical pipe, and the outside diameter and thickness are estimated by counting and size estimation [24]. The detailed process is described in Section 2.2, with a schematic diagram.

### 2.2. Overlapping Segmentation Labeling for Pipe Size Estimation

The CNN-based image-processing techniques, devised from supervised learning through the usage of datasets prepared in advance to address objects in an image or an image itself, can be divided into three main categories: image classification; object detection, which expresses the minimum area that includes an object in an image as rectangular coordinates; and instance segmentation, which expresses the boundaries of an object as polygon coordinates. The duration and computing costs of the image processing conducted in CNN modeling differ largely, depending on the purpose and the CNN architecture; thus, it is important to use an appropriate CNN model that matches the purpose of use.

As the pixels occupied by the cross-section of a pipe in an image are directly related to the diameter and thickness of the pipe, the exact pixel that occupies the cross-section must be segmented. The cross-section of a pipe consists of an outside circle that serves as the basis for the outside diameter, and an inside circle that serves as the basis for the inside diameter; moreover, the thickness refers to the difference between the outside and inside diameters. Therefore, to infer the thickness of a pipe from an image, an overlapping segmentation was applied in this study to simultaneously extract the outside and inside circles from a single pipe.

To train a CNN model that can segment the outside and inside circles of a single pipe into different polygons by using instance segmentation, three types of images were collected, as shown in Figure 2: a single pipe, a single name tag, and multiple pipe piles with name tags. The outside and inside circles of a pipe were annotated by overlapping the outer and inner labels with several polygon points. The name tag was annotated as four points by the label of the tag. The pipe circle that becomes covered by the name tag in the multiple pipe piles entailing the name tags was annotated by cropping the section that becomes covered.

The images that were used to form the dataset were collected as full high-definition sizes with pixel sizes of 1920 × 1080. Then, they were cropped to a size of 1080 × 1080, relative to the center, and were subsequently down-sampled to a size of 550 × 550 to increase the computing speed. After pre-processing, the dataset containing 536 images of the pipe with a total of 3652 pipe cross-sections and 235 name tags was established for the segmentation of the outside and inside circles, and the name tags in the images. In the dataset, the number of pipe cross-sections that were photographed in a single image varied from 1 to 30, and a 0–1 name tag was included. The annotation tool, Labelme [25], was used to assign the polygon ground-truth bounding boxes to the pipe cross-sections. The image was photographed at various angles, and up to four images were taken for the same pipe or pipe pile.

### 2.3. Pipe Diameter and Thickness Estimation Process

The detailed process of the estimation and counting of the pipe’s diameter and thickness after segmenting the outside and inside circles, and the name tag in the CNN model, are shown in Figure 3. The name tag that was identified in the segmentation image had a transformation matrix that was generated by the calculation of the corresponding points in the actual scale of the front-view direction applied to the entire image, indicating that a virtual image is generated in the front-view direction by a homography. At this point, the same transformation matrix was also applied to the polygon that composes the segmentation, and the virtual image consisted of displacement coordinates of the actual scale. Furthermore, the actual dimensions of the name tag used in this study were 9 × 6.5 cm, and corner detection was applied to the name tag that was identified by the CNN model.

If multiple pipes are present in a single image, a matching process is conducted to match the outside and inside circles of the same pipe. The matching recognizes two circles with the closest coordinate distance between the center points of the outside and inside circles as circles of the same pipe. The Euclidean distance between the two points (the centers of the outside and inside circles) is calculated by Equation (1):(1)$$d_{ij}=\|center_{oc}^{i}-center_{ic}^{j}\|$$
where, $center_{oc}^{i}$ and $center_{ic}^{j}$ refer to the center coordinates of the *i*-th outside circle and the *j*-th inside circle, respectively, and $d_{ij}$ refers to the Euclidean distance between the *i*-th outside circle and the *j*-th inside circle. The same pipe, with respect to the *i*-th outside circle, is calculated by Equation (2):(2)$$pipe_{i}=\min\left(d_{ij}\right).$$

For a single pipe whose outside and inside circles matched, the equations for calculating the outside diameter and the thickness from the area of the outside and inside circles are as follows:(3)$$Outside\;diameter=\sqrt{\frac{4\times Area_{oc}}{\pi }},\;Inside\;diameter=\sqrt{\frac{4\times Area_{ic}}{\pi }}$$
where, $Area_{oc}$ and $Area_{ic}$ refer to the area of the outside and inside circles, respectively. By deriving the diameter from the area, we can derive the value that can be acquired from averaging the value of the diameter measured from all angles, and this compensates for the errors that occur during segmentation. The thickness of the pipe can be calculated by the difference between the outside and inside diameters:(4)$$Thickness=\frac{Outside\;diameter-Inside\;diameter}{2}.$$

## 3. Results

### 3.1. Splitting Datasets Using K-Fold Cross-Validation

In this section, we present the research results. The hardware for the algorithm testing was composed of the NVIDIA GeForce RTX 2080 Ti Graphics Processing Unit (GPU). YOLACT [26] was used to train a CNN model that can segment objects in an image. Transfer learning was conducted based on the pre-trained weights of the coco dataset [27] to enhance the accuracy of the model.

The K-fold cross-validations were conducted to classify the collected datasets into the training and test datasets prior to generating the model through training. In general, the K-fold cross-validations are used to eliminate the data consumed by validation when data is scarce, to increase the number of datasets used in training, and to subsequently enhance the accuracy and prevent underfitting. In this study, we compared the mean average precision (mAP) of the 10 folds that randomly divided 536 images into training and test datasets at the ratios of 70:30, 80:20, and 90:10, after training each fold 10,000 times. The training results of the top 9 and the total mean of the 30 generated folds are shown in Table 1. As the segmentation polygons of the outside and inner circles are used for pipe counting and size estimation, the mAP of the mask from the 1000th, 5000th, and 10,000th training sessions was compared. In most cases, the accuracy did not considerably improve after the 5000th training. The highest mAP and total mean mAP were observed from the third fold that was divided at the ratio of 80:20. Thus, the fold was ultimately used as a dataset for the generation of a CNN model.

### 3.2. Training and Evaluation

The datasets were divided into 429 and 107 pages for the training and tests, respectively, and had data argumentation according to the image processing applied to generate additional six datasets for each image, in which 3003 training images and 749 test images were used in the final training. Figure 4 depicts the box loss, mask loss, the box mAP, and mask mAP of the test dataset that was not used for training as a result of the training progression. In the case of the mask mAP—the core data of this study—the accuracy was drastically enhanced until the 3000th training and it gradually increased until approximately 12,000 epochs, where it did not increase further. A model entailing a mask mAP of 44.8 and a box mAP of 47.7 was ultimately acquired at 15,000 epochs.

### 3.3. Results of the Samples

The results of the two sample images on the estimation and counting of the pipe diameter and thickness in the test data are portrayed in Figure 5. The raw image collected for each image, the homography image on the transformation matrix of the segmentation model and the name poly tag, and the matching image resulting from the Euclidean distance, are illustrated in this order. The outside diameter and thickness that show the results of the size estimation of the pipe for the matching images are shown as bar plots, and the errors of the estimated result, with respect to the actual size, are also listed.

Figure 5a depicts a partial photograph of 19 NPS5 pipes with a diameter of 141.3 mm and a thickness of 6.553 mm, and one NPS24 pipe with a diameter of 609.6 mm and a thickness of 9.525 mm. The outside and the inside circles became overlapped in the segmentation and homography images, respectively, meaning that 20 of each were detected. In the matching image, 20 pipes were matched with the inside circle based on the outside circle. In the case of the NPS5 pipe, the outside diameter of the fourth pipe was estimated to be 152.4 mm, with a maximum error of 7.8%. In the case of the NPS22 pipe, the actual outside diameter was 609.6 mm, however, the diameter was estimated to be 168.31 mm because part of the image was cut off, resulting in an error of 72.4%. The thickness of the fourth pipe of NPS5 was estimated to be 5.69 mm with the largest error of 13.2%, and all the pipes had errors of less than 1 mm. In particular, in the case of the NPS24 pipe, the incision of the image considerably affected the estimation of the outside diameter, but it did not affect the estimation of the thickness, which is calculated by the difference between the areas of the outside and inside circles, resulting in a small error of 2.5%.

Figure 5b depicts a photograph of 26 NPS4 pipes with a diameter of 114.3 mm and a thickness of 7.137 mm, and two NPS$\frac{1}{2}$ pipes with a diameter of 21.3 mm and a thickness of 2.413 mm. A total of 28 outside and inside circles were detected in the segmentation and homography images. In the matching image, 28 pipes were matched with the inside circle based on the outside circle. A maximum error of 64% in the outside diameter estimation occurred in the incised image contained the 1st, 2nd, 8th, and 25th NPS4 pipe and the 14th NPS$\frac{1}{2}$2 pipe. Although an error of more than 20% occurred, errors smaller than 15% were obtained in all the pipes while estimating the thickness values.

## 4. Conclusions

In this paper, a computer vision-based pipe quantification technique is presented by calculating the diameter and thickness of pipes from stacked-pipe images. The proposed technique consists of three steps: a CNN model generation, an image and coordinate transformation, and dimension estimation. A CNN model, capable of segmenting the outside and inside circles of pipes, is trained from various images. The application of the homography transformation to the segmented coordinates of the pipe circles that were generated from the CNN model generates a virtual front-view image on a real scale. The labeling of individual pipes and their thickness calculation is conducted by the Euclidean distance matching proposed in the study.

For the CNN model generation, 536 images of a 550 × 550 size, including 3652 pipes and 235 name tags, were prepared and labeled. After YOLACT was adopted as the CNN model for segmentation and the 10,000 K-fold cross-validations were performed, a CNN model was created by training it 15,000 times on the optimal dataset. Precise accuracy verification was performed on two sample images. Except for the case where the image of the pipe was partially cut off, the outside diameter error was evaluated up to 7.8%. A less than 15% error of thickness estimation was confirmed independently of the cropped images. In future studies, it is suggested to consider the application of a CNN model capable of high-resolution segmentation to improve the accuracy of the thickness calculation.

## Figures and Tables

**Figure 1 sensors-22-04517-f001:**
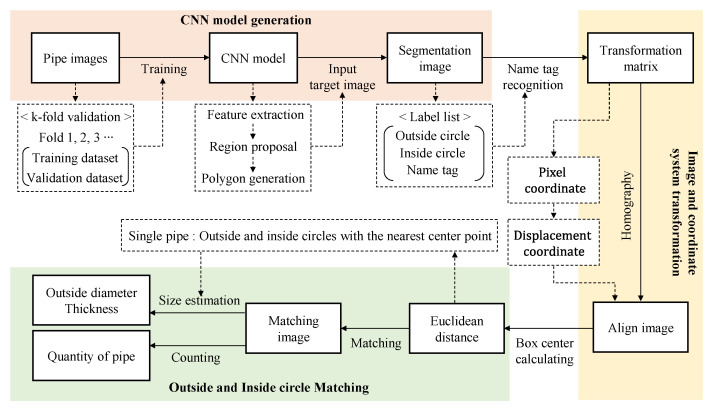
Flow chart for estimating the number and size of steel piles.

**Figure 2 sensors-22-04517-f002:**
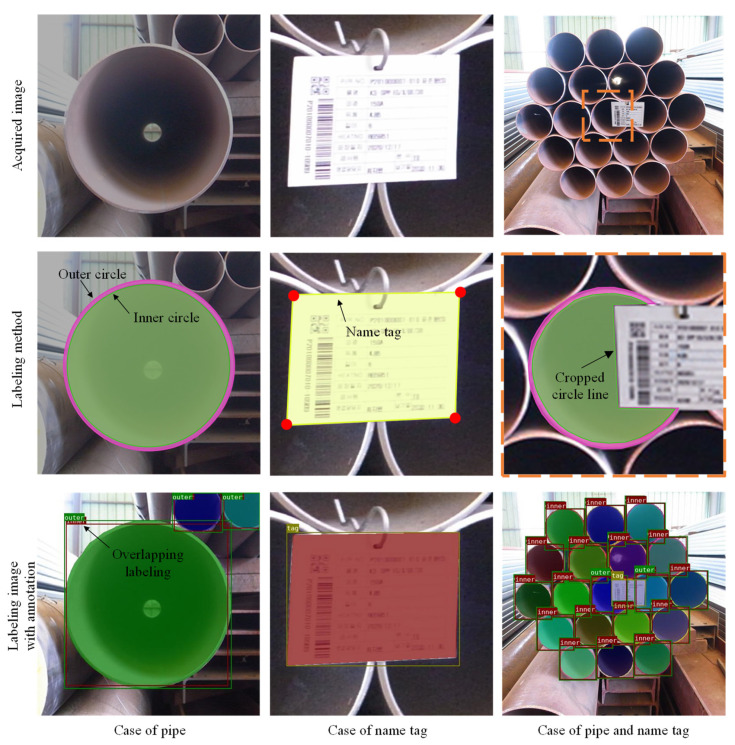
Labeling method for representative images with annotations.

**Figure 3 sensors-22-04517-f003:**
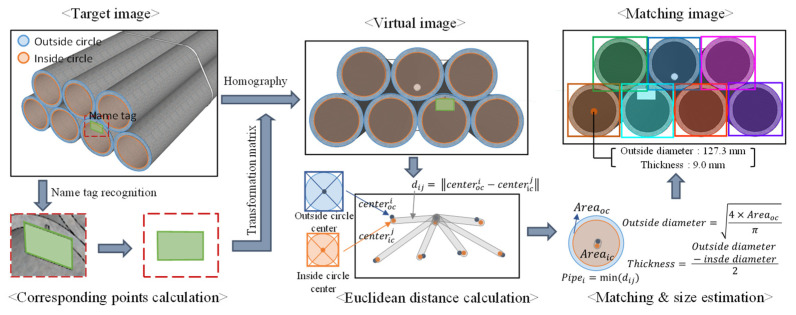
Schematic diagram of pipe counting and size estimation.

**Figure 4 sensors-22-04517-f004:**
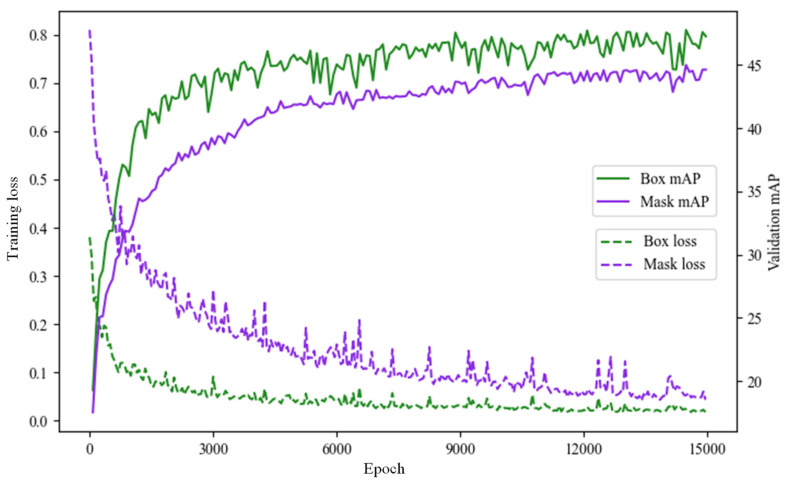
Total loss and mAP of box and mask by training.

**Figure 5 sensors-22-04517-f005:**
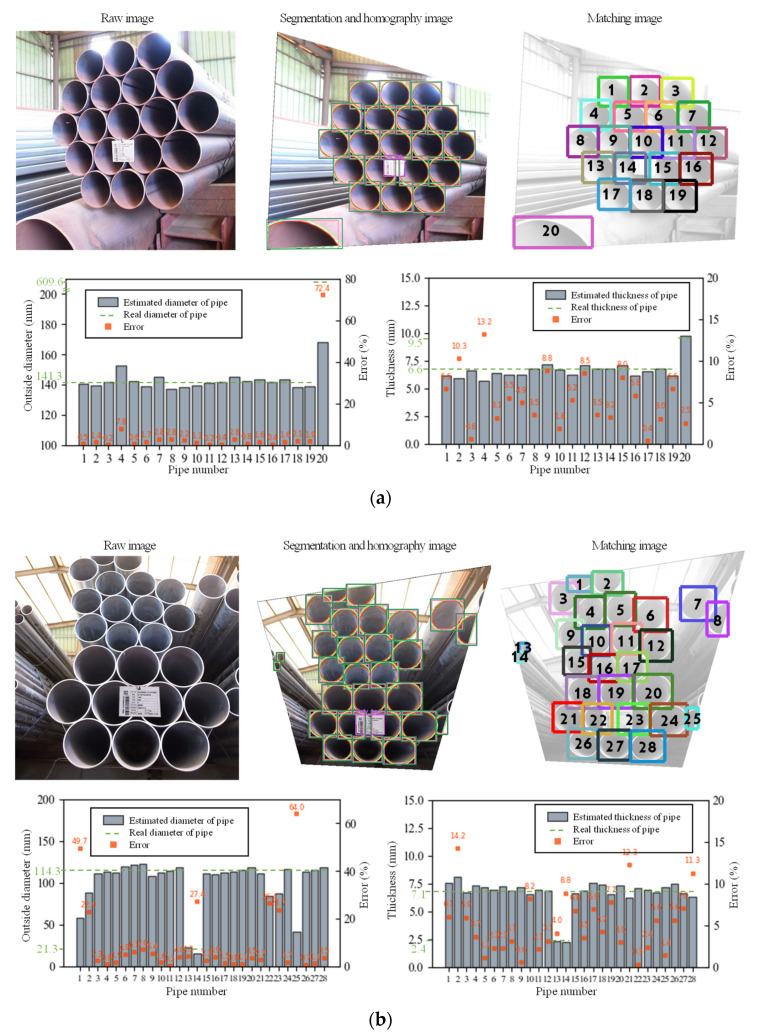
Pipe size estimation process and results. (**a**) Sample 1, (**b**) Sample 2.

**Table 1 sensors-22-04517-t001:** Part of K-fold cross-validation result.

Split Ratio (Training: Set)	mAP of Mask													
	Fold	Label	Outside			Inside			Name Tag			All		
		Iteration	1k	5k	10k	1k	5k	10k	1k	5k	10k	1k	5k	10k
70:30	1		33.4	39.2	41.1	31.7	40	40.9	32.8	38.3	41.1	32.6	39.2	41.1
	2		32.9	41.5	39.5	33.5	41.6	38.7	35.1	40.7	41.1	33.8	41.3	39.8
	3		32.5	38.8	37.3	32.2	40.5	39.8	33.7	39.4	37.8	32.8	39.6	38.3
	Total mean		29.1	38.4	37.8	28.7	36.9	37.6	31	36.5	39.8	29.6	37.3	38.4
80:20	1		34.1	42.2	43.3	34.6	41.5	43.9	34.5	38.9	45.7	34.4	40.9	44.3
	2		32.5	40.1	42.2	33.9	39.6	42	34.1	40.8	41.3	33.5	40.2	41.8
	3		31.6	38.1	38.6	33.7	36.8	39	31.7	35.5	38.9	32.3	37.1	38.8
	Total mean		31.9	39.9	37.3	32.1	38.8	40.4	32.3	34.2	40.2	32.1	37.6	39.3
90:10	1		33.1	40.2	41.3	31.7	39	41.5	30.8	37	39.2	31.9	38.7	40.7
	2		32.1	36.8	39.1	32.1	36.4	40.3	34.4	37.7	41.2	32.9	37	40.2
	3		31.1	36.1	38.3	29.5	34	37.2	30.9	35.2	35.5	30.5	35.1	37
	Total mean		28.8	33.7	39.2	27.2	31.6	37.3	30	33.1	35.7	28.7	32.8	37.4

## Data Availability

The data used to support the results of this study are included within the article. Furthermore, some of the data in this research are supported by the references mentioned in the article. If you have any queries regarding the data, the data of this research will be available from the correspondence upon request.

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
