# Peer review of "Automated Dimension Estimation of Steel Pipes Stacked at Construction Sites Using Euclidean Distances Calculated by Overlapping Segmentation"

_sensors, 2022, doi:10.3390/s22124517_

Round 1

Reviewer 1 Report

-please add performance obtained also to the abstract to make it more complete 

-line 13 is not clear

-CNN models are created is not scientifically sounding. Please rephrase 

-there are some typos and sytactic mistakes in the abstract, and overall in the paper. Edit it under the point of view of the english and semantic correctness 

-lin 57: CNN generates a neural network is not scientifically correct. Moreover there is a bit of confusion in the introduction between the parts where related material is given and the parts in which instead the CNN is described. MY suggestion is to give more space also to the CNN parts (you can also decide to divide into 2 paragrpahs). 

Relevant papers concerning CNN and deep learning should be cited. For example: 

  -Havaei, Mohammad, et al. "Brain tumor segmentation with deep neural networks." Medical image analysis 35 (2017): 18-31.

-Jégou, Simon, et al. "The one hundred layers tiramisu: Fully convolutional densenets for semantic segmentation." Proceedings of the IEEE conference on computer vision and pattern recognition workshops. 2017

-Bianchini, Monica, et al. "Deep neural networks for structured data." Computational Intelligence for Pattern Recognition. Springer, Cham, 2018. 29-51

-LeCun, Yann, Yoshua Bengio, and Geoffrey Hinton. "Deep learning." nature 521.7553 (2015): 436-444

-Did you try other distance measures apart from the Euclidean? This choice should at least be discussed in the paper 

-Add description to caption of Figure 1

-Please add details in the text of all of the variables/notations introduced in equatsions from 1 onwards

-Add more details and description to Figure 3 

-generate a CNN model is not semantically correct as an expression. Use a different expression (implement or train or test when relevant

-Table 1 only shows 9 out of 30 folds. Could you please add all of the results? At least in the supplementary material they should be added

- Did you use a validation set to optimize hyperparameters?

-Howe many epochs were used_ At line 231 you are mentioning 12000 which looks like definetely too many considering the number of samples available. The risk of overfitting in this case is high

Reviewer 2 Report

The paper presents an end-to-end system for performing automated pipe counting and size identification. The author use the appropriate technologies that have existed for a couple of years in order to perform the task in question.

The presented results show that they are quite successful. The paper is generally well written with good command of English.

The reviewer requests two things:

a) a public github with the developed code in order to ensure the reproducibility of the presented research.

b) there is no comparison with other approaches/methods on the subject. As a research paper, this comparison is always necessary in order to demonstrate how we have improved to previous approaches. This is not present in the paper and must be included.
